# Japonamides A and B, Two New Cyclohexadepsipeptides from the Marine-Sponge-Derived Fungus *Aspergillus* *japonicus* and Their Synergistic Antifungal Activities

**DOI:** 10.3390/jof8101058

**Published:** 2022-10-09

**Authors:** Haifeng Wang, Rui Zhang, Ben Ma, Wenzhao Wang, Chong Yu, Junjie Han, Lingjuan Zhu, Xue Zhang, Huanqin Dai, Hongwei Liu, Baosong Chen

**Affiliations:** 1School of Traditional Chinese Materia Medica, Shenyang Pharmaceutical University, Shenyang 110016, China; 2State Key Laboratory of Mycology, Institute of Microbiology, Chinese Academy of Sciences, Beijing 100190, China

**Keywords:** cyclohexadepsipeptides, synergistic antifungal activities, *Aspergillus japonicus*, marine-sponge-derived fungus, japonamides

## Abstract

Two new cyclohexadepsipeptides japonamides A (1) and B (2) were isolated from the ethyl acetate extract of a marine-sponge-derived fungus *Aspergillus japonicus* based on molecular networking. Their structures were elucidated by comprehensive spectral analysis and their absolute configurations were confirmed by Marfey’s method. Compounds 1 and 2 showed no antifungal activities against *Candida albicans* SC5314 measured by the broth microdilution method but exhibited prominent synergistic antifungal activities in combination with fluconazole, ketoconazole, or rapamycin. The Minimum inhibitory concentrations (MICs) of rapamycin, fluconazole, and ketoconazole were significantly decreased from 0.5 to 0.002 μM, from 0.25 to 0.063 μM, and from 0.016 to 0.002 μM, in the presence of compounds 1 or 2 at 3.125 μM, 12.5 μM, and 6.25 μM, respectively. Surprisingly, the combination of compounds 1 or 2 with rapamycin showed a strong synergistic effect, with fractional inhibitory concentration index (FICI) values of 0.03.

## 1. Introduction

The destruction of the immune system makes immunocompromised people vulnerable to fungal infections, such as those suffering from AIDS, cancer after chemotherapy, etc. [1,2,3]. Fungal infection is one of the main causes of death in these patients [4]. In recent years, fungi, such as *Candida albicans* and *Aspergillus fumigatus*, led by invasive fungi, have gradually become important pathogens leading to nosocomial infection [5]. The clinical infection and mortality rates have increased year by year due to fungal infection [6]. *Candida albicans*, one of the most common opportunistic pathogens in the genus *Candida*, has the highest infection rate and mortality [7]. At present, although there are many antifungal drugs with certain effects, an increasing number of cases of clinical treatment failure have occurred, mainly due to the emergence of drug-resistant strains and new strains of *Candida albicans* [8]. Drug combination is an effective strategy to overcome drug resistance. The combination of antifungal drugs and non-antifungal drugs can achieve synergy by overcoming drug resistance or enhancing antifungal drug activity [9,10]. The strategy of drug combination is widely adopted in clinical practice. Therefore, the discovery of efficient natural products with synergistic effects with antimicrobial agents has become a hot spot to solve the problem of antibiotic resistance.

Cyclic peptides (depsipeptides) are a kind of cyclic compound mainly formed by amino acids linked by peptide bonds. They are widely distributed, ranging from bacteria and fungi to higher plants and mammals. Great success has been achieved in the development of cyclic peptide drugs [11,12,13]. The classical cyclic peptide drugs include cyclosporine, vasopressin, vancomycin, and oxytocin. Compared with other small-molecule drugs, the structural properties of cyclic peptides contribute to form orderly secondary structures, prevent potential targeting side effects, and produce harmless metabolites. Many cyclic peptides or depsipeptides were reported to have antifungal or synergistic antifungal activities, and are potential antifungal drugs or lead compounds; for example, tunicyclin D extracted from *Psammosilene tunicoides,* showed a broad spectrum of antifungal activity against *Candida* genus [14]. Westertides A and B from *Aspergillus westerdijkiae* showed synergistic antifungal activity against *Candida albicans* SC5314 with the presence of rapamycin [15]. Cyclopentapeptides from an Endolichenic *Xylaria* sp. showed synergistic antifungal activity against *Candida albicans* SC5314 with ketoconazole [16]. Therefore, new cyclic peptides are urgently needed to enrich the library of antifungal lead compounds.

Traditional strategies for mining novel cyclic peptides were not attractive for the repetitive and redundancy discovery of known compounds. Cyclic peptides and depsipeptides often produce characteristic mass fragments for the amide and/or ester bonds in the structure that are susceptible to cleavage during collisions in mass spectrometers [17]. Molecular networking can automatically group compounds with similar fragmentation patterns according to the tandem mass data and make them visualizable [18]. As a result, MS/MS-based molecular networking is a highly efficient strategy for the discovery of peptide natural products. In our long-term study on marine fungi, a marine-sponge-derived fungus *Aspergillus japonicus* was selected to explore cyclic peptides based on molecular networking, and a small number of metabolites such as insecticidal activity compounds paraherquamide A [19] and asperparaline A [20] were reported in this fungus. This exploration led to the discovery of two new cyclohexadepsipeptides with synergistic anti-*Candida albicans* effect, namely japonamides A (1) and B (2). The details of isolation, structural identification, and biological activities are described herein.

## 2. Materials and Methods

### 2.1. General Experimental Procedures

Optical rotations were determined on MCP200 with a 1 dm length cell at 25 °C. UV spectra were recorded by Shimadzu UV-1700 UV spectrometer (Shimadzu, Tokyo, Japan). NMR spectral data were measured with a Bruker AVANCE-500 spectrometer (Bruker, Karlsruhe, Germany) (DMSO-*d*_6_, *δ*_H_ 2.50/*δ*_C_ 39.52). IR spectra were taken on a Bruker IFS-55 infrared spectrophotometer (Bruker, Karlsruhe, Germany). CD spectra were recorded with a BioLogic MOS-450 spectrometer (BioLogic Science, Grenoble, France). LC-MS/MS data were collected from Agilent Accurate-Mass-Q-TOF LC/MS 6520. Semipreparative HPLC was performed on an Agilent 1200 HPLC system equipped with an Agilent DAD UV−vis spectrometric detector, using a reversed-phase Eclipse XDB-C18 column (5 μm, 9.4 mm × 250 mm, Agilent) with a flow rate of 2.0 mL/min. Silica gel (Qingdao Haiyang Chemical Co., Ltd., 200–300 mesh), Sephadex LH-20 (Pharmacia, Uppsala, Sweden), and ODS (50 μm, YMC CO., LTD) were used for column chromatography. Fractions were monitored by TLC (silica gel GF254, Qingdao Marine Chemical Co. Ltd.), and the spots were visualized by UV at 254 nm and spraying with 10% H_2_SO_4_-EtOH assisted with heating. All reagents or solvents were HPLC or analytical grade and were purchased from Tianjin Damao Chemical Company (Tianjin, China) unless otherwise stated.

### 2.2. Fungal Material, Fermentation, and Extraction

The fungus *Aspergillus japonicus* was purchased from the Marine Culture Collection of China (MCCC 3A00261) which was isolated from the marine sponge collected from the Arctic 6700-4 sea area and identified according to the morphological analysis and ITS gene sequencing (Genbank Accession number HM573340). The *Aspergillus japonicus* strain was cultured on slants of Potato Dextrose Agar (PDA) medium at 28 °C for 7 days as seed medium. For large-scale fermentation, agar plugs were inoculated in 309 bottles of 500 mL Erlenmeyer flasks with 200 mL of PDA medium and then incubated at 25 °C for 35 days under static cultivation.

After fermentation, the mycelium and fermentation broth were separated by suction filtration. The 60 L fermentation broth was concentrated to 5 L, and the same amount of ethyl acetate was added, after extracting 3 times to obtain 10 g ethyl acetate extract under reduced pressure.

### 2.3. Molecular Networking Analyses and Compound Isolation

An amount of 10 mg ethyl acetate extract was dissolved in 2 mL of methanol and analyzed by LC-MS/MS. The resulting raw data were converted to mzML format and analyzed using the Molecular Networking tool on the Global Natural Product Social (GNPS) Web site (https://gnps.ucsd.edu/ProteoSAFe/static/gnps-splash.jsp, accessed on 22 April 2022). A mass tolerance of 0.02 Da was set for both the precursor ion and the MS/MS fragment ion. The minimum pairs’ cosine, matched fragment ions, network topk, maximum connected component size, and cluster size were set to 0.7, 6, 10, 100, and 2, respectively. The results were downloaded and visualized with Cytoscape 3.9.1 [17,21].

The 10 g EtOAc extract was fractionated by vacuum liquid chromatography on silica gel (200–300 mesh) using CH_2_Cl_2_/CH_3_OH gradient elution (100:1–0:100, *v*/*v*) to give five fractions, Fr.1–Fr.5. Guided by molecular networking, peptides were found in Fr.3. Fr.3 (4.14 g) was subjected to reversed-phased ODS column chromatography (CH_3_OH−H_2_O, 10−80%, *v*/*v*) and obtained 9 subfractions (Fr.3-1 to Fr.3-9). Fr.3-5 (50 mg) was purified by semipreparative reversed-phase HPLC with 2.0 mL/min 60% CH_3_OH−H_2_O to provide japonamide A (1, 8.8 mg, t_R_ = 22.8 min) and japonamide B (2, 6.5 mg, t_R_ = 34.2 min).

Japonamide A (1): white powder, [*α*]25D = −68.0 (c 0.1, MeOH); UV (MeOH) *λ*_max_ (log *ε*) = 210 (4.55), 220 (sh) nm; CD (c = 0.5 mg/mL, MeOH) *λ*_max_ (∆ε) = 233 (−38) nm; IR (neat) *ν*_max_ = 3415, 3295, 2972, 2968, 2936, 1670, 1623, 1522, 1454, 1246, 1067, 1027, 828, 806, 737 cm^−1^; ^1^H and ^13^C NMR data, see Table 1. Positive HRESIMS *m/z* [M + Na]^+^ 827.3951 (calcd. for C_42_H_56_N_6_O_10_Na, 827.3956).

Japonamide B (2): white powder, [*α*]25D = −120.98 (c 0.1, MeOH); UV (MeOH) *λ*_max_ (log *ε*) = 210 (4.53), 220 (sh) nm; CD (c = 0.5 mg/mL, MeOH) *λ*_max_ (∆ε) = 233 (−32) nm; IR (neat) *ν*_max_ = 3415, 3289, 2963, 2968, 2936, 1628, 1514, 1454, 1246, 1067, 1027, 828, 806, 737, 701 cm^−1^; ^1^H and ^13^C NMR data, see Table 1. Positive HRESIMS *m/z* [M + Na]^+^ 813.3800 (calcd. for C_41_H_54_N_6_O_10_Na, 813.3799).

### 2.4. Absolute Configurations of Amino Acids by the Advanced Marfey’s Analysis

The advanced Marfey’s analyses were carried out as previous reported with some modifications [16,21]. Compounds 1 and 2 (2.0 mg) were dissolved in 6 N HCl (2.0 mL) and heated at 100 °C for 24 h. The solutions were then evaporated to dryness and transferred to a 4 mL reaction vial and treated with a 10 mg/mL solution of 1-fluoro-2-4-dinitrophenyl-5-L-alanine amide (FDAA, 200 μL) in acetone, followed by 1.0 M NaHCO_3_ (40 μL). The reaction mixtures were heated at 45 °C for 90 min, and the reactions were quenched by the addition of HCl (1 N, 40 µL). Similarly, standard l- and d-amino acids (Thr, Val, Pro, Tyr and *N*-Me-Tyr-OMe) were derivatized separately. The derivatives of the acid hydrolysate and the standard amino acids were subjected to HPLC analysis (Kromasil C18 column; 5 μm, 4.6 × 250 mm; 1.0 mL/min; UV detection at 340 nm) with a linear gradient of acetonitrile (30–45%) in water (TFA, 0.01%) over 50 min. Retention times for the authentic standards were as follows: l-Thr (5.9 min), d-Thr (6.9 min), l-Pro (8.8 min), d-Pro (9.6 min), FDAA (11.0 min), l-Val (14.2 min), *N*-Me-*O*-Me-l-Tyr (19.3 min), l-Ile (19.9 min), d-Val (20.5 min), *N*-Me-*O*-Me-d-Tyr (21.2 min), d-Ile (28.1 min), l-Tyr (36.3 min), and d-Tyr (43.7 min). The absolute configurations of the chiral amino acids in compounds 1 and 2 were determined by comparing the retention times.

### 2.5. In Vitro Activities of Compounds 1 and 2 in Combination with Antibiotics against Candida albicans *SC5314*

The strain used for antifungal and synergistic antifungal bioassay was *Candida albicans* SC5314. Antifungal susceptibility testing was carried out as described previously [16,22] in 96-well microtiter plates (Greiner, Germany), using a broth microdilution protocol modified from the Clinical and Laboratory Standards Institute M-27A3 methods. The concentrations were 2-fold diluted from 100 to 1.56 μM (test compounds) or from 1 to 0.0156 μM (positive drugs). Minimum inhibitory concentration (MIC) was determined as the drug concentration that inhibits fungal growth by >80% relative to the corresponding drug-free growth control. For the synergistic antifungal testing, 1/4 MIC of one compound was preadded into the medium, with the procedures being otherwise carried out in the same fashion. Basic procedures were the same as the method for antifungal susceptibility testing. Antifungal agents (rapamycin, fluconazole, and ketoconazole) were 2-fold diluted from the concentrations 2 to 0.002 μM in column, while peptide-like compounds were 2-fold diluted from 25 to 0.39 μM in row of the 96-well microtiter plate. The fractional inhibitory concentration index (FICI) is defined as the sum of the MIC of each drug when used in combination divided by the MIC of the drug used alone. Synergism and antagonism were defined by FICI indices of ≤0.5 and >4, respectively.

## 3. Results and Discussion

After extraction and concentration, 10 g ethyl acetate extract was gained from 60 L PDA fermentation broth of *Aspergillus japonicus*. The EtOAc extract was analyzed by high-resolution tandem mass spectrometry (HR-MS/MS). A molecular network (MN) was constructed using the GNPS Molecular Networking platform (Figure 1A). More than 15 prominent clusters were observed. Upon inspection, the annotated network revealed the presence of a particularly interesting “molecular family” constituted of nodes reminiscent of peptides (Figure 1B), as the tandem mass spectra of these protonated molecules featured typical fragments corresponding to amino acid imine ions (Figure 1C). Guided by the MN, compounds in this cluster were emphatically isolated, and two new cyclohexadepsipeptides japonamides A (1) and B (2) were obtained (Figure 2). These structures were determined by extensive spectroscopic analysis and Marfey’s reaction.

Japonamide A (1) was isolated as white powder. The molecular formula C_42_H_56_N_6_O_10_ of compound 1 was deduced from a positive HR-ESI-MS ion at *m/z* 827.3951 (calcd. for C_42_H_56_N_6_O_10_Na, 827.3956), with 18 degrees of unsaturation. In the IR spectrum, the stretching vibration of N-H (*v*_N-H_3415 cm^−1^), O-H (*v*_O-H_ 3295 cm^−1^), C-H (*v*_C-H_ 2972, 2968, 2936 cm^−1^), C=O (*v*_C=O_ 1623 cm^−1^), C=C of aromatic group (*v*_Ar_-_C=C_ 1670, 1522 cm^−1^), and C-N (*v*_C=C_ 1454 cm^−1^) indicated the existence of NH, OH, C=O, and a benzene ring. Analysis of its ^1^H, ^13^C, and HSQC NMR data (Table 1, Appendix A) indicated the presence of three amide N-proton signals (*δ*_H_ 8.66 (d, *J* = 8.7 Hz), 7.78 (d, *J* = 9.1 Hz), and 7.35 (d, *J* = 8.9 Hz)) and eight methine groups, seven of which oxygenated or nitrogenated at *δ*_C_/_H_ 69.9/4.96, 61.8/4.79, 57.3/4.62, 55.7/4.25, 54.5/4.71, 54.5/4.31, and 54.5/4.42, seven carbonyl carbons at *δ*_C_ 172.2, 170.7, 170.0, 169.8, 169.6, 169.3, and 167.7, two 1,4-substituted benzene rings at *δ*_C_/_H_ 130.4/7.10 (2H, d, *J* = 8.9 Hz), 113.9/6.85 (2H, d, *J* = 8.9 Hz), 130.1/6.95 (2H, d, *J* = 8.5 Hz), 115.0/6.65 (2H, d, *J* = 8.5 Hz), and *δ*_C_ 127.8, 129.8, 155.9, and 158.1, five methyl groups including one methoxy at *δ*_C_/_H_ 55.1/3.70, one doublet methyl at *δ*_C_/_H_ 14.7/0.82 (d, *J* = 6.7 Hz), one triplet methyl at *δ*_C_/_H_ 11.6/0.85 (*t*, *J* = 7.4 Hz), another two singlet methyl groups at *δ*_C_/_H_ 28.2/2.23 and 22.7/1.98, and nine methylene groups. All of the above information implies the existence of peptide or depsipeptide.

The ^1^H-^1^H COSY correlations (Figure 3, Appendix A) H_2_-3 with H_2_-4 and H-2, and H_2_-4 with H-5, and HMBC correlations (Figure 3, Appendix A) from H_2_-3 and H-2 to C-1 elucidated the structure of proline residue. Another proline residue was constructed with the same structural analysis method. The ^1^H-^1^H COSY correlations of H-12 with H_2_-13, and H-15/19 with H-16/18, combined with HMBC correlations from OCH_3_ protons (*δ*_H_ 3.70) to C-17, from H-16/18 to C-17 and C-14, from H-15/19 to C-13, C-14 and C-17, from H-12 and H2-13 to C-11, and from NCH_3_ protons (*δ*_H_ 2.23) to C-12, identified the N-methyl-O-methyl tyrosine (diMeTyr) residue. Similarly, the tyrosine residue was confirmed. The isoleucine residue was illustrated by the ^1^H-^1^H COSY correlations of H-39 with H-38, H_2_-40 and H_3_-41, and H_2_-40 and H_3_-43, together by the HMBC correlation from H-39 and H-38 to C-37. The ^1^H-^1^H COSY correlations of H-33 with H-32 and H_3_-34, the HMBC correlations from H-33 and H-32 to C-31, and the chemical shift of oxygenated methine-33 (*δ*_C_/_H_ 69.9/4.96) together confirmed the threonine residue. Ulteriorly, the HMBC correlations from H_3_-36 and H-32 to C-35 (*δ*_C_ 170.0) verified threonine residue was substituted by acetyl and it was an N-acetylthreonine residue (AcThr).

The AcThr-Ile-Pro-Pro-diMeTyr-Tyr sequential connection of amino acid residues was joined by the HMBC correlations from NH proton (*δ*_H_ 7.35) and H-40 to C-31, the ROESY correlations (Figure 3, Appendix A) of H_2_-5 with H-38, and H_2_-10 with H-2, the HMBC correlations from NCH_3_ protons (*δ*_H_ 2.23) and H-12 to C-6, and from NH proton (*δ*_H_ 8.66) and H-23 to C-11. The peptide chain was closed by the HMBC correlations from H-33 to C-22. Geometry was present in the proline amide bond, and the Δ*δ*_Cβ_-_Cγ_ values of the Pro residues (3.5 and 3.0 ppm for Pro1 and Pro2, respectively) were indicative of trans geometries for all proline amide bonds in 1 [23,24].

Furthermore, to elucidate the structure of the cyclohexadepsipeptide undoubtedly, HR-MS/MS was also used to determine the amino acid sequence. According to the mechanism of amide bond cleavage, protonated peptides in the low collision energy regime will generate sequence-diagnostic ion series containing the N-terminus (b_n_ fragments), C-terminus (y_n_ fragments), and a fragment (b_n_ losing CO) [25,26,27]. As shown in Figure 1 and Figure 4, the first dissociation step of compound 1 involves the opening of the cyclo-peptide ring by cleavage of the lactone group and results in a linear form fragment. One possible fragment (b_6_) ion losing Tyr residue, H_2_O, diMeTyr residue, Pro residue, Pro residue, and Ile residue, successively, generated the fragments b_5_ (642), b_5_ (624), b_4_ (433), b_3_ (336), b_2_ (239), and b_1_ (126). Another possible fragment ion losing AcThr, Ile, Pro, and Pro residues generated the fragments y_5_ (680), y_4_ (567), y_3_ (470), and y_2_ (373), successively. In addition, other abundance ions were deduced in Figure 1. The analysis of HR-MS/MS matched with the NMR, and the amino acid sequence of cyclohexadepsipeptide was confirmed undoubtedly.

The absolute configurations of the amino acid moieties in compound 1 were deciphered using the advanced Marfey’s analysis, following HPLC comparison against available l and d commercial standards [17]. Acid hydrolysis (HCl) of 1 and chemical derivatization with FDAA yielded a mixture of FDAA derivative of Thr (derived from *N*-Ac-l-Thr), Ile, Pro, *N*-Me-*O*-Me-Tyr, and Tyr. HPLC analyses of the mixture of hydrolysates and appropriate amino acid standards confirmed the presence of *N*-Ac-l-Thr, d-Ile, l-Pro (×2), *N*-Me-*O*-Me-l-Tyr, and l-Tyr in 1 (Figure 5, Appendix A). Thus, compound 1 was identified and named as japonamide A.

Japonamide B (2) was isolated as white powder (methanol). The molecular formula C_41_H_54_N_6_O_10_ was deduced from a positive HR-ESI-MS ion at *m/z* [M+Na]^+^ 813.3800 (calcd. for 813.3799) with 18 degrees of unsaturation. The ^1^H NMR spectroscopic data showed that compound 2 was similar to compound 1. One triplet methyl (CH_3_-42) and one methylene (CH_2_-40) in 1 replaced by one doublet methyl (*δ*_H_ 0.86, CH_3_-41) in 2 indicated that isoleucine residue in compound 1 was replaced by valine residue in 2. HMBC correlations from *δ*_H_ 7.40 (NH) to *δ*_C_ 55.8 (C-38) and *δ*_C_ 29.8 (C-39), from *δ*_H_ 4.32 (H-38) to C-39, *δ*_C_ 18.6 (C-41), and *δ*_C_ 18.7 (C-40), from *δ*_H_ 0.85 (H_3_-40) and *δ*_H_ 0.86 (H_3_-41) to C-39 and C-38, proved to existence of valine in 2. Comprehensive analysis of NMR (Figure 3, Appendix A), HR-MS/MS (Figure 4), and Marfey’s reaction (Figure 5, Appendix A) resulted in the structure of compound 2 being assigned as cyclo-[(*N*-Me-*O*-Me-l-Tyr)-l-Pro-l-Pro-d-Val-(*N*-Ac-l-Thr)- l-Tyr] and named as japonamide B.

Many cyclic peptides or depsipeptides were reported to have antifungal or synergistic antifungal activities against the azole-resistant strain *Candida albicans* SC5314 in the presence of rapamycin or ketoconazole [14,15,16]. In this work, the antifungal or synergistic antifungal activities against *Candida albicans* SC5314 were evaluated. Compounds 1 and 2 showed no antifungal activities at all by themselves (MICs > 100 μg/mL). Checkerboard assays were used to obtain the minimum synergistic concentrations with rapamycin, fluconazole, and ketoconazole (Table 2). The MICs of rapamycin, fluconazole, and ketoconazole were significantly decreased from 0.5 to 0.002 μM, from 0.25 to 0.063 μM, and from 0.016 to 0.002 μM, respectively, while in the presence of compounds 1 or 2 at 3.125 μM, 12.5 μM, and 6.25 μM, respectively. The FICIs lower than 0.5 indicated that compounds 1 and 2 had synergistic antifungal effects with rapamycin, fluconazole, and ketoconazole. In addition, compounds 1 and 2 showed more efficient synergistic antifungal activity with rapamycin. Moreover, cytotoxicity was evaluated, and neither two compounds showed cytotoxicity within 100 μM against RAW264.7, U973, HT22, and PC12 cell lines.

## 4. Conclusions

Two new cyclohexadepsipeptides were isolated from the ethyl acetate extract of a marine-sponge-derived fungus based on molecular networking. Their structures were elucidated by spectroscopic analysis, and their absolute configurations were confirmed by Marfey’s method. All compounds showed synergistic effects with antifungal drugs (fluconazole, ketoconazole, and rapamycin). Compounds 1 and 2 combined with rapamycin revealed synergistic antifungal activity against *Candida albicans*, with MIC values of 3.125 μM and FICI values of 0.03. The absence of toxicity to mammalian cells indicated the safety of the drug, which has important implications for the research and development of drugs. The results of the present study suggest that the combination of compounds 1 or 2 with antifungal drugs may be an effective anti-*Candida albicans* regimen. Meanwhile, the results opened up a new method for cyclic peptides as synergistic antifungal active molecules in combination with fluconazole, ketoconazole, and rapamycin against resistant *Candida albicans*. The underlying synergistic mechanism requires further exploration.

## Data Availability

Not applicable.

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
