# Peer review of "Japonamides A and B, Two New Cyclohexadepsipeptides from the Marine-Sponge-Derived Fungus Aspergillus japonicus and Their Synergistic Antifungal Activities"

_jof, 2022, doi:10.3390/jof8101058_

Round 1
Reviewer 1 Report
The manuscript by Wang et al. describes the isolation and structure analysis of two cyclohexadepsipeptides with antifungal activities. I will not comment on the sections dealing with the extraction and molecular analyses of these peptides, as these are outside my area of expertise. My review will be limited to the parts relating to in vitro antifungal susceptibility and interaction testing. In this regard, the analyses done was quite limited, as it included only a single fungal species/isolate, and interaction with just 2 classes of molecules.
Comments
1. Introduction: the references in this section do not match or support the statements in the text. For example, Ref 4, which is cited after a statement regarding the mortality rate of candidiasis, is a survey of fluconazole susceptibility in Candida isolates and contains no clinical data. I was not able to locate Refs 5 and 6 in Pubmed.
The introduction contains some vague statements with no supporting references: see lines 26-27, 29-30, 32-33.
2. It is unclear why the authors decided to test interaction with 3 specific molecules: 2 azole drugs and rapamycin. The peptides could have been tested with representatives of all three main systemic antifungal classes, as well as a panel of non-antifungal molecules, including CNI, cell wall active agents, protein kinase inhibitors, etc.
3. The authors found no significant activity of the cyclic peptides alone against Candida albicans SC5314. More information about the spectrum of activity of the cyclic peptides would be helpful. e.g. activity against other Candida species and medically important filamentous fungi.
4. Synergistic activity was found for the combination of cyclic peptides and fluconazole, ketoconazole and rapamycin. Was this activity fungistatic or cidal? In other words, could the C. albicans cells from peptide-drug combination wells be cultured on drug-free medium?
5. Toxicity of cyclic peptides against mammalian cells should be tested, to clarify the potential of these molecules as novel antifungal drug leads.
6. “Super-hyper synergistic effect” (Abstract) is not an accepted term. Please use instead “strong” or “potent” synergistic effect.
Author Response
Dear reviewers,
Thanks for your kind comments and suggestions for improving our manuscript. We have made a careful revision on our manuscript according to the comments. All the changes made in manuscript are highlighted. Our answers to comments and suggestion are listed as follows.
We highly appreciate your hard work in reviewing and editing this submission. Any of further questions and suggestions are welcomed.
Sincerely yours,
Dr. Hongwei Liu
State Key Lab of Mycology, Institute of Microbiology, CAS
- Introduction: the references in this section do not match or support the statements in the text. For example, Ref 4, which is cited after a statement regarding the mortality rate of candidiasis, is a survey of fluconazole susceptibility in Candida isolates and contains no clinical data. I was not able to locate Refs 5 and 6 in Pubmed.
The introduction contains some vague statements with no supporting references: see lines 26-27, 29-30, 32-33.
Reply: Thanks for your suggestions. We have checked the references and modified them, and added the references in the manuscript for the statements (lines 26-27, 29-30, and 32-33). At the same time, references 5 could be located through the following website links.
Refs 5 (Ref 7 in the revised version): https://doi.org/10.1016/B978-0-12-809633-8.12075-8
- It is unclear why the authors decided to test interaction with 3 specific molecules: 2 azole drugs and rapamycin. The peptides could have been tested with representatives of all three main systemic antifungal classes, as well as a panel of non-antifungal molecules, including CNI, cell wall active agents, protein kinase inhibitors, etc.
Reply: Candida albicans SC5314 is a clinical strain with azoles resistance. It was sensitive to other antifungal classes. In our previous work, we reported the synergistic antifungal activities of many cyclic peptides or depsipeptides against Candida albicans SC5314 with the presence of immunosuppressor rapamycin. Thus, we chose the rapamycin, fluconazole and ketoconazole.
- The authors found no significant activity of the cyclic peptides alone against Candida albicans SC5314. More information about the spectrum of activity of the cyclic peptides would be helpful. e.g. activity against other Candida species and medically important filamentous fungi.
Reply: The compounds were used for structure elucidation, and bioassay. Residual compounds were too little too test other bioactivities.
- Synergistic activity was found for the combination of cyclic peptides and fluconazole, ketoconazole and rapamycin. Was this activity fungistatic or cidal? In other words, could the C. albicans cells from peptide-drug combination wells be cultured on drug-free medium?
Reply: Synergistic activity was fungicidal, C. albicans from peptide-drug combination wells can’t be cultured on drug-free medium.
- Toxicity of cyclic peptides against mammalian cells should be tested, to clarify the potential of these molecules as novel antifungal drug leads.
Reply: we test the cytotoxicity, the two compounds showed no toxicity within 100 μM against RAW264.7, U973, HT22 and PC12 cell lines.
- “Super-hyper synergistic effect” (Abstract) is not an accepted term. Please use instead “strong” or “potent” synergistic effect.
Reply: We have replaced the “Super-hyper” with the “strong” in the abstract.
Reviewer 2 Report
The MS entitled `` New cyclohexadepsipeptides from the Marine Sponge-Derived Fungus Aspergillus japonicas and their synergistic antifungal activities`` cannot be published in the current form. The following issues should be addressed:
The title should be modified to Japonamides A and B New cyclohexadepsipeptides from the Marine Sponge-Derived Fungus Aspergillus japonicas and their synergistic antifungal activities
Write the full name of all abbreviations when they first appear.
The results of antifungal activity for the compounds and control as well as the type of assay should be added in the abstract
Add the compounds name `` Japonamides`` in the keywords
English editing is mandatory, there are many grammatical and typing mistakes.
The genus name should be italicized throughout the whole MS
An introduction about the previously reported metabolites from this fungus as well as their bioactivities should be added
The city, country, and manufacturer should be added for each instrument in the experimental
The fungal GenBank Accession number should be added from
Some instruments are missing IR and CD
The IR spectral data should be discussed in the MS
The concentration of tested compounds should be added.
In figure 3, the authors should specify to which each arrow and bold lines refer
What the authors measured here is ROESY not NOE, correct in the MS.
The role of ROESY in structure elucidation should be mentioned in detail.
The listed IR data and UV data should be discussed along with their significance in structure characterization.
Check all coupling constant values in the table, they should be one decimal.
Authors should mention the reason behind the evaluation antifungal potential of these metabolites.
Author Response
Dear reviewers,
Thanks for your kind comments and suggestions for improving our manuscript. We have made a careful revision on our manuscript according to the comments. All the changes made in manuscript are highlighted. Our answers to comments and suggestion are listed as follows.
We highly appreciate your hard work in reviewing and editing this submission. Any of further questions and suggestions are welcomed.
Sincerely yours,
Dr. Hongwei Liu
State Key Lab of Mycology, Institute of Microbiology, CAS
- The title should be modified to Japonamides A and B New cyclohexadepsipeptides from the Marine Sponge-Derived Fungus Aspergillus japonicas and their synergistic antifungal activities
Reply: we change the title as, “Japonamides A and B, two new cyclohexadepsipeptides from the marine sponge-derived fungus Aspergillus japonicus and their synergistic antifungal activities”
- Write the full name of all abbreviations when they first appear.
Reply: We checked and corrected the whole manuscript.
- The results of antifungal activity for the compounds and control as well as the type of assay should be added in the abstract
Reply: it has been added in abstract.
- Add the compounds name `` Japonamides`` in the keywords
Reply: We have added the ``japonamides`` in the keywords.
- English editing is mandatory, there are many grammatical and typing mistakes.
Reply: We apologize for the poor language of our manuscript. We worked on the manuscript for a long time and the repeated addition and removal of sentences and sections obviously led to poor readability.
- The genus name should be italicized throughout the whole MS
Reply: We have checked and corrected them in the manuscript.
- An introduction about the previously reported metabolites from this fungus as well as their bioactivities should be added.
Reply: According to your advice, we have added the previously reported metabolites from this fungus as well as their bioactivities.
- The city, country, and manufacturer should be added for each instrument in the experimental
Reply: We have supplemented relevant information.
- The fungal GenBank Accession number should be added.
Reply: We have added the fungal GenBank Accession number in the section of Fungal Material, Fermentation and Extraction.
- Some instruments are missing IR and CD
Reply: We have added the instruments of IR and CD in the section of General Experimental Procedures.
- The IR spectral data should be discussed in the MS
Reply: IR data and UV data have been discussed in MS.
- The concentration of tested compounds should be added.
Reply: The concentration of tested compounds has been added in experiment part.
- In figure 3, the authors should specify to which each arrow and bold lines refer.
Reply: The explanation of arrow and bold lines have been added.
- What the authors measured here is ROESY not NOE, correct in the MS.
Reply: it has been corrected.
- The role of ROESY in structure elucidation should be mentioned in detail.
Reply: Rotating Frame Overhauser Effect Spectroscopy (ROESY) is one of common 2D method to express the NOE effects. ROESY correlation signals indicated the two protons were closest in space. In my opinion, just like HMBC, HSQC, and 1H-1H COSY, we do not have to explain the common methods in detail.
- The listed IR data and UV data should be discussed along with their significance in structure characterization.
Reply: IR data have been discussed in MS.
- Check all coupling constant values in the table, they should be one decimal.
Reply: it has been corrected.
- Authors should mention the reason behind the evaluation antifungal potential of these metabolites.
Reply: We describe the reason in introduction indeed. This time we describe the reason in result, briefly.
Round 2
Reviewer 1 Report
Reference 7 is cited incorrectly.
Please refer to the Journal's instructions on citing a book chapter.
First author is Yanez, A. Book title is "Encyclopedia of Mycology", unabbreviated. Include names of editors, publisher and place published.
Details can be found here: https://www.elsevier.com/books/encyclopedia-of-mycology/zaragoza/978-0-12-819990-9
In some references, author first names are listed in stead of last names: e.g. Ref 4, Ref 5 - please correct.
Author Response
Reference 7 is cited incorrectly. Please refer to the Journal's instructions on citing a book chapter. First author is Yanez, A. Book title is "Encyclopedia of Mycology", unabbreviated. Include names of editors, publisher and place published. Details can be found here: https://www.elsevier.com/books/encyclopedia-of-mycology/zaragoza/978-0-12-819990-9
In some references, author first names are listed in stead of last names: e.g. Ref 4, Ref 5 - please correct.
Reply: the reference 7 have been changed as book refernce style. And the authors’ name of reference 4-7 have been changed.
Reviewer 2 Report
The MS can be accepted. But the English needs refining.
Author Response
The MS can be accepted. But the English needs refining.
Reply: thanks for your kind suggestion. This time we do some change in the MS.